# New Antimalarial and Antimicrobial Tryptamine Derivatives from the Marine Sponge *Fascaplysinopsis reticulata*

**DOI:** 10.3390/md17030167

**Published:** 2019-03-15

**Authors:** Pierre-Eric Campos, Emmanuel Pichon, Céline Moriou, Patricia Clerc, Rozenn Trépos, Michel Frederich, Nicole De Voogd, Claire Hellio, Anne Gauvin-Bialecki, Ali Al-Mourabit

**Affiliations:** 1Laboratoire de Chimie des Substances Naturelles et des Sciences des Aliments, Faculté des Sciences et Technologies, Université de La Réunion, 15 Avenue René Cassin, CS 92003, 97744 Saint-Denis CEDEX 9, La Réunion, France; pierre-eric.campos@univ-reunion.fr (P.-E.C.); pichon.manu@orange.fr (E.P.); patricia.clerc@univ-reunion.fr (P.C.); 2Institut de Chimie des Substances Naturelles, CNRS UPR 2301, Univ. Paris-Sud, Université Paris-Saclay, 1, av. de la Terrasse, 91198 Gif-sur-Yvette, France; Celine.Moriou@cnrs.fr (C.M.); Ali.ALMOURABIT@cnrs.fr (A.A.-M.); 3Laboratoire des Sciences de l’Environnement MARin (LEMAR), Université de Brest, CNRS, IRD, Ifremer, LEMAR, F-29280 Plouzane, France; rozenn.trepos@port.ac.uk (R.T.); Claire.Hellio@univ-brest.fr (C.H.); 4Laboratory of Pharmacognosy, Center for Interdisciplinary Research on Medicines, CIRM, University of Liège B36, 4000 Liège, Belgium; M.Frederich@ulg.ac.be; 5Naturalis Biodiversity Center, Darwinweg 2, 2333 CR Leiden, The Netherlands; nicole.devoogd@naturalis.nl

**Keywords:** *Fascaplysinopsis reticulata*, marine sponge, tryptamine alkaloids, antimalarial activity, antimicrobial activity

## Abstract

Chemical study of the CH_2_Cl_2_-MeOH (1:1) extract of the sponge *Fascaplysinopsis reticulata* collected in Mayotte highlighted three new tryptophan derived alkaloids, 6,6′-bis-(debromo)-gelliusine F (**1**), 6-bromo-8,1′-dihydro-isoplysin A (**2**) and 5,6-dibromo-8,1′-dihydro-isoplysin A (**3**), along with the synthetically known 8-oxo-tryptamine (**4**) and the three known molecules from the same family, tryptamine (**5**), (*E*)-6-bromo-2′-demethyl-3′-*N*-methylaplysinopsin (**6**) and (*Z*)-6-bromo-2′-demethyl-3′-*N*-methylaplysinopsin (**7**). Their structures were elucidated by 1D and 2D NMR spectra and HRESIMS data. All compounds were evaluated for their antimicrobial and their antiplasmodial activities. Regarding antimicrobial activities, the best compounds are (**2**) and (**3**), with minimum inhibitory concentration (MIC) of 0.01 and 1 µg/mL, respectively, towards *Vibrio natrigens*, and (**5**), with MIC values of 1 µg/mL towards *Vibrio carchariae*. In addition the known 8-oxo-tryptamine (**4**) and the mixture of the (*E*)-6-bromo-2′-demethyl-3′-*N*-methylaplysinopsin (**6**) and (*Z*)-6-bromo-2′-demethyl-3′-*N*-methylaplysinopsin (**7**) showed moderate antiplasmodial activity against *Plasmodium falciparum* with IC_50_ values of 8.8 and 8.0 µg/mL, respectively.

## 1. Introduction

Tryptophan-derived alkaloids are well-established bioactive metabolites and have been isolated from various marine organisms: sponges, scleratinian corals, one sea anemone and one nudibranch [1]. Species of the sponge genus *Fascaplysinopsis* have yielded several bioactive tryptophan alkaloids reported to exhibit cytotoxic activity against several cancer cell lines [2,3], antimicrobial [2], antiviral [4] and antimalarial [5] activities.

In our continuing search for bioactive metabolites from marine invertebrates, the sponge *Fascaplysinopsis reticulata* (Hentschel, 1912) from the Dictyoceratida order was investigated. Previous studies on *Fascaplysinopsis reticulata* collected from the Benga Lagoon of the Fiji Islands by Jiménez et al. [6], and then from Indonesia (Molucca Sea) and from the Fiji Islands by Segraves et al. [7], led to the isolation of 23 alkaloids from the fascaplysin family. More recent study on *Fascaplysinopsis reticulata* collected from Xisha Island (China) by Wang et al. led to the isolation of a pair of bisheterocyclic quinolineimidazole alkaloids, (+)- and (−)-spiroreticulatine [8]. All of the isolated 25 molecules are tryptophane-derived alkaloids.

Our chemical investigation of the extract of *Fascaplysinopsis reticulata* collected in Mayotte (Indian Ocean), led to the isolation of three new members of the tryptophan family, 6,6′-bis-(débromo)-gelliusine F (**1**), 6-bromo-8,1′-dihydro-isoplysin A (**2**) and 5,6-dibromo-8,1′-dihydro-isoplysin A (**3**), along with the known derivatives 8-oxo-tryptamine (**4**), tryptamine (**5**), (*E*)-6-bromo-2′-demethyl-3′-*N*-methylaplysinopsin (**6**) and (*Z*)-6-bromo-2′-demethyl-3′-*N*-methylaplysinopsin (**7**). The 8-oxo-tryptamine (**4**) was known as synthetic compound [9], but was isolated here from a natural source. We report herein the purification and structure elucidation by spectral data including HRESIMS, 2D NMR and comparison with published data. The biological evaluations of the latter new compounds are described as well.

## 2. Results and Discussion

### 2.1. Chemistry

The CH_2_Cl_2_-MeOH extract of the lyophilized sponge *Fascaplysinopsis reticulata* was first subjected to a reverse-phase silica gel column chromatography to yield fractions. The fractions were subjected to repetitive reverse-phase semi-preparative and analytical HPLC to yield eight compounds (**1**–**7**) (Figure 1). Three were new: one 6,6′-bis-(debromo)-gelliusine F (**1**) and two aplysinopsin derivatives **2** and **3**, described below. In addition to the new compounds, four other known members were identified as 8-oxo-tryptamine (**4**), tryptamine (**5**) and a mixture of (*E*)-6-bromo-2′-demethyl-3′-*N*-methylaplysinopsin (**6**) and (*Z*)-6-bromo-2′-demethyl-3′-*N*-methylaplysinopsin (**7**) by comparison with published spectroscopic data.

6,6′-bis-(debromo)-gelliusine F (**1**) was obtained as a brown oil. The molecular formula, C_20_H_23_N_4_, was established from HRESIMS molecular ion peak at *m*/*z* 319.2013 [M + H]^+^. Analysis of the 1D and 2D ^1^H, and ^13^C NMR data for **1** (CD_3_OD, Table 1) revealed resonances and correlations (Figure 2) consistent with those of a bis-tryptamine structure linked by the carbons C-2 and C-8′, like gelliusine F [10,11]. Analysis of the HSQC correlations and the comparison with latter compounds pointed the fragment C-8, C-9, C-9′ (δH 3.23, 3.00, 3.83–3.69; δC 23.7, 41.4, 44.3), one aliphatic methine C-8′ (δH 5.10; δC 34.3), nine aromatic methines C-4, C-5, C-6, C-7, C-2′, C-4′, C-5′, C-6′, C-7′ (δH 7.54, 7.38, 7.12, 7.06, 7.27, 7.58, 7.41, 7.14, 7.06; δC 118.9, 112.6, 123.1, 120.7, 124.0, 119.3, 112.9, 123.3, 120.6) and seven nonprotonated aromatic carbons C-2, C-3, C-3a, C-7a, C-3′, C-3a’, C-7a’ (δC 124.0, 113.8, 127.5, 135.3, 113.7, 129.2, 138.0). Compound **1** was different from gelliusine F by the presence of the two aromatic methines C-6 and C-6′ instead of two nonprotonated aromatic carbons substituted by bromine. Analysis of the COSY correlations revealed the presence of the spin systems C-4−C-5−C-6−C-7 and C-4′−C-5′−C-6′−C-7′ and confirmed this difference. These COSY correlations, in addition to the HMBC correlations between H-4 and C-7a, between H-5 and C-3a, between H-6 and C-7a, between H-2′, C-3′, C-3a’ and C-7a’, between H-4′ and C-7a’, between H-5′ and C-3a’, between H-6′ and C-7a’ and between H-7′ and C-3a’, confirmed the presence of two indole cores, the first one substituted in C-2 and C-3 and the second one substituted in C-3′. The COSY correlation between H-8 and H-9 and the HMBC correlation between H-9 and C-3 and between H-8, C-2, C-3 and C-3a indicated the substitution of the first indole core by an ethylamine chain in C-3. The COSY correlation between H-8′ and H-9′ and the HMBC correlations between H-9′ and C-3′ and between H-8′, C-2′, C-3′ and C-3a’ indicated the substitution of the second indole core by an ethylamine chain in C-3′. The two tryptamine patterns were linked between C-2 and C-8′ like gelliusine F. Compound **1** was named 6,6′-bis-(debromo)-gelliusine F according to gelliusine F, reported in 1995 [11].

6-bromo-8,1′-dihydro-isoplysin A (**2**) was obtained as a yellow oil. Its molecular formula, C_14_H_16_BrN_4_O (9 degree of unsaturation), was established from HRESIMS pseudo-molecular ion peak at *m*/*z* 337.0483 (see Appendix A) indicating the presence of one bromine atom in the molecule. Analysis of the 1D and 2D ^1^H, and ^13^C NMR data for **2** (CD_3_OD, Table 2) revealed resonances and correlations (Figure 2) consistent with those of a 1′,8-dihydroaplysinopsin structure: the HSQC correlations revealed the presence of one methylene C-8 (δH 3.35; δC 28.1), one aliphatic methine C-1′ (δH 4.62; δC 61.8), four aromatic methines C-2, C-4, C-5, C-7 (δH 7.11, 7.51, 7.14, 7.50; δC 126.4, 121.1, 123.3, 115.5), four nonprotonated aromatic carbons C-3, C-3a, C-6, C-7a (δC 109.0, 127.6, 116.6, 138.1), one guanidine-like carbon C-3′ (δC 159.2) and one amide carbonyl C-5′ (δC 174.9). The structure of the indole core was determined by the analysis of COSY correlations between H-4 and H-5, the ^4^*J* coupling constant between H-5 and H-7 (J = 1.8 Hz) and HMBC correlations between H-2, C-3, C-3a, and C-7a, between H-4, C-6 and C-7a and between H-5, C-3a and C-7. The HMBC correlation between H-2 and C-8 indicated the substitution of the non-protonated carbon C-3 by the methylene C-8. The COSY correlation between H-8 and H-1′ indicated link between the heterocycle core and C-8. The structure of the heterocycle core was determined by the HMBC correlations between H-1′ and C-5′, between CH_3_-6′ and C-3′ and between CH_3_-7′, C-3′ and C-5′. 

5,6-dibromo-8,1′-dihydro-isoplysin A (**3**) was obtained as a yellow oil. Its molecular formula C_14_H_15_Br_2_N_4_O (9 degrees of unsaturation), was established from HRESIMS pseudo-molecular ion peak at *m*/*z* 414.9630 (see Appendix A) indicating the presence of two bromine atom in the molecule. Analysis of the ^1^H and ^13^C NMR data for **3** and comparison with the ^1^H and ^13^C NMR data for **2** (CD_3_OD, Table 3) revealed a 1′,8-dihydroaplysinopsin structure close to the above-described 6-bromo-8,1′-dihydro-isoplysin A (**2**), where one hydrogen was replaced by a bromine atom. The spectra showed two *N*-methyles C-6′, C-7′ (δH 2.86, 2.94; δC 25.4, 28.9), one methylene C-8 (δH 3.73; δC 28.2), one aliphatic methine C-1′ (δH 4.60; δC 61.4), three aromatic methines C-2, C-4, C-7 (δH 7.16, 7.96, 7.69; δC 126.6, 123.6, 117.4), five nonprotonated aromatic carbons C-3, C-3a, C-5, C-6, C-7a (δC 109.0, 129.8, 116.9, 115.9, 137.5), one guanidine-like carbon C-3′ (δC 157.9) and one amide carbonyl C-5′ (δC 175.8). 5,6-dibromo-8,1′-dihydro-isoplysin A (**3**) differed from 6-bromo-8,1′-dihydro-isoplysin A (**2**) by the presence of one more aromatic nonprotonated aromatic carbon and one less aromatic methine. The chemical shifts and the multiplicity of C-4 and C-7 also differed between compound **3** (two singlets) and compound **2** (two doublets). For compound **3**, the multiplicity of C-4 and C-7 indicated that H-4 was *para* to H-7. These spectroscopic features, as well as the molecular formula, supported that the position of the proton H-5 of **2** was substituted by a bromine in compound **3**.

### 2.2. Microfouling Activity

The capacity of compounds to interfere with microfouling was assessed by screening the pure compounds against five bacterial strains that are involved in the initial formation of the fouling biofilm: *Shewanella putrefaciens, Roseobacter litoralis, Vibrio carchariae, Vibrio natrigens* and *Vibrio proteolyticus.* The effects on both adhesion and growth (A and G) were studied, and the results expressed as the minimal inhibitory concentration (MIC) are summarized in Table 4. The two new 6-bromo-8,1′-dihydro-isoplysin A (**2**) and 5,6-dibromo-8,1′-dihydro-isoplysin A (**3**) showed promising antifouling activity against *Vibrio natrigens*, with MIC values of 0.01 and 1.00 µg/mL, respectively, towards growth inhibition. *Vibrio natrigens* is a major component of biofilms due to its fast generation doubling time, its biofilm producing ability and steel corrosion behavior. Thus, it has considerable negative economic impacts on man-made immersed surfaces [12,13].

The activity of these compounds was lower towards inhibition of adhesion (respectively 100 and >100 µg/mL for (**2**) and (**3**)). Based on MICs values and mode of action, 6-bromo-8,1′-dihydro-isoplysin A (**2**) is the most potent compound as it has the ability to reduce growth when used at very low concentration and can also affect adhesion at higher doses. Regarding the known compound, tryptamine (**5**) showed promising antimicrobial activity against *Vibrio carchariae* with MIC value of 1 µg/mL. This result is of high interest, as *Vibrio carchariae* is responsible for mass mortalities of fish [14] and invertebrates [15]. Thus, *Vibrio carchariae* is considered to be a major nuisance for the aquaculture sector [16], and new ways to stop its development are sought after.

### 2.3. Antiplasmodial Activity

All the isolated compounds were also tested against the protozoan parasite *Plasmodium falciparum* (3D7 strain). The 8-oxo-tryptamine (**4**) and the mixture of the known (*E*) and (*Z*)-6-bromo-2′-demethyl-3′-*N*-methylaplysinopsin (**6**, **7**) exhibited antiplasmodial activity against *Plasmodium falciparum* with IC_50_ values of 8.8 and 8.0 µg/mL respectively while 6,6′-bis-(debromo)-gelliusine F (**1**), 6-bromo-8,1′-dihydro-isoplysin A (**2**), 5,6-dibromo-8,1′-dihydro-isoplysin A (**3**) and tryptamine (**5**) did not show significant antimalarial activity. Hu et al. [17] have already reported the antiplasmodial activity of (*E*) and (*Z*)-6-bromo-2′-demethyl-3′-*N*-methylaplysinopsin (**6**, **7**) together with the activity of two other aplysinopsins, isoplysin A and 6-bromoaplysinopsin isolated from the sponge *Smenospongia aurea*. Bialonska et Zjawiony [1] also reported, for 27 aplysinopsins, their biological activities, among which the antiplasmodial activity seems to be dependent on the skeleton: all the aplysinopsins that presented antiplasmodial activity had a double bond between C-8 and C-1′. The lack of antiplasmodial activity for compounds (**2**) and (**3**) confirms this study. These activities are moderate compared to control drugs, but these simple molecular scaffolds could be investigated for further pharmacomodulations in order to improve final bioactivity. 

## 3. Materials and Methods

### 3.1. General Experiment Procedures

Optical rotations were measured on a MCP 300 polarimeter (Anton Paar, Les Ulis, France) at 25 °C (MeOH, *c* in g/100 mL). ^1^H and ^13^C NMR data were acquired with a Brucker UltraShield Avance-300 and 600 MHz spectrometers (CNRS-ICSN, Brucker, Wissembourg, France). Chemical shifts were referenced using the corresponding solvent signals (δ_H_ 3.31 and δ_C_ 49.00 for CD_3_OD). The spectra were processed using TopSpin software (TopSpin 3.5, Brucker, Wissembourg, France). HRESIMS spectra were recorded using a Waters Acquity BEH C18, 1.7 μm, 50 × 2.1 mm column on a Waters Micromass LCT-Premier TOF mass spectrometer (Waters, Guyancourt, France) with a Waters Acquity UPLC system.

The sponge was lyophilized with Cosmos −80 °C CRYOTEC and extracted with Dionex ASE 300. Reversed phase column chromatography separations were carried out on glass column (150 × 10 mm i.d.) packed with Acros Organics C18-RP, 23%C, silica gel (40−63 μm). Precoated TLC sheets of silica gel 60, Alugram SIL G/UV254 were used, and spots were visualized on the basis of the UV absorbance at 254 nm and by heating silica gel plates sprayed with formaldehyde−sulfuric acid or Dragendorff reagents. Analytical HPLC was carried out using a Waters Sunfire C_18_ (150 × 4.6 mm i.d., 5 μm) column and was performed on an Agilent 1100 series system controller equipped with a photodiode array detector (Serie Agilent 1100 G1315B, Agilent Technologies, Wilmington, Germany) and a mass spectrometer detector (Serie Agilent 1100 G1956A, Agilent Technologies, Wilmington, Germany) with Chemstation software (Version B.04.03. Agilent Technologies, Wilmington, Germany). Preparative HPLC was carried out using a Waters Sunfire Prep RP_18_ (150 × 19 mm i.d., 5 μm) column and was performed on a Waters 600 system controller equipped with a photodiode array detector (Waters 2996, Waters, Guyancourt, France). Semi-preparative HPLC was carried out using Waters Sunfire Prep RP_18_ (250 × 10 mm i.d., 5 μm) column and was performed on a Waters 600 system controller (Waters, Guyancourt, France) equipped with photodiode array detectors (Waters 2996 and Waters 486). All solvents were analytical or HPLC grade and were used without further purification.

### 3.2. Animal Material

The sponge *Fascaplysinopsis reticulata* (phylum Porifera, class Demospongiae, order Dictyoceratida, family Thorectidae) was collected in May 2013 in Passe Bateau, Mayotte (12°58,653′ S, 44°58,949′ E at 15–17 m depth). One voucher specimen (RMNH POR 8466) was deposited in Naturalis, the Netherlands Centre for Biodiversity. Sponge samples were frozen immediately and kept at −20 °C until processed.

### 3.3. Extraction and Isolation

The frozen sponge (28 g, dry weight) was chopped into small pieces and extracted by ASE first with Water (×1) and then with MeOH/CH_2_Cl_2_ (1:1, *v:v*) (×2). After evaporating the solvents under reduced pressure, a brown, oily residue (2.91 g) was obtained. The extract (2.90 g) was then subjected to fractionation by C-18 SPE, eluted with a combination of Water, MeOH and CH_2_Cl_2_ of decreasing polarity and twelve fractions were obtained (F1–F12).

Fraction F3 (543 mg). Separation of only 100 mg of this fraction was performed by preparative HPLC (Waters Sunfire Prep C_18_ Column, 5 µm, 150 × 19 mm i.d., 18 mL min^−1^ gradient elution with 2% ACN-H_2_O (+0.1% formic acid) over 5 min, then 10% ACN-H_2_O (+0.1% formic acid) to 100% ACN over 30 min; UV 280 nm) to furnish pure compound **1** (6,6′-bis-(debromo)-gelliusine F, 0.6 mg).

Fraction F4 (355.4 mg). Only 200 mg was subjected to preparative HPLC (Waters Sunfire Prep C_18_ Column, 5 µm, 150 × 19 mm i.d., 18 mL min^−1^ gradient elution with 2% ACN-H_2_O (+0.1% formic acid) over 5 min, then 2% ACN-H_2_O (+0.1% formic acid) to 100% ACN (+0.1% formic acid) over 35 min; UV 280 nm) to give pure compounds **2** (6-bromo-8,1′-dihydro-isoplysin A, 4 mg), **3** (5,6-dibromo-8,1′-dihydro-isoplysin A, 4 mg) and **5** (tryptamine, 4.0 mg).

Fraction F5 (99.1 mg) was subjected to preparative HPLC (Waters Sunfire Prep C_18_ Column, 5 µm, 150 × 19 mm i.d., 18 mL min^−1^ gradient elution with 2% ACN-H_2_O (+0.1% formic acid) over 5 min, then 2% ACN-H_2_O (+0.1% formic acid) to 100% ACN (+0.1% formic acid) over 45 min; UV 280 nm) to give pure compound **1** (6,6′-bis-(debromo)-gelliusine F, 1.5 mg), **4** (8-oxo-tryptamine, 0.7 mg) and **5** (tryptamine, 3.0 mg).

Fraction F6 (51.1 mg) was subjected to semi-preparative HPLC (Waters Sunfire Prep RP_18_ Column, 5 μm, 250 × 10 mm i.d., 4.5 mL min^−1^ gradient elution with 2% ACN-H_2_O (+0.1% formic acid) over 5 min, then 2% ACN-H_2_O (+0.1% formic acid) to 100% ACN (+0.1% formic acid) over 35 min; UV 280 nm) to give pure compounds **2** (6-bromo-8,1′-dihydro-isoplysin A, 1.2 mg), **4** (8-oxo-tryptamine, 0.4 mg) and **5** (tryptamine, 0.6 mg).

Fraction F7 (266.8 mg) was subjected to semi-preparative HPLC (Waters Sunfire Prep RP_18_ Column, 5 μm, 250 × 10 mm i.d., 4.5 mL min^−1^ gradient elution with 2% ACN-H_2_O (+0.1% formic acid) over 5 min, then 2% ACN-H_2_O (+0.1% formic acid) to 100% ACN (+0.1% formic acid) over 35 min; UV 280 nm) to give pure compounds **5** (tryptamine, 0.4 mg) and the mixture of **6** and **7** ((*E*) and (*Z*)-6-bromo-2′-demethyl-3′-*N*-methylaplysinopsin, 10 mg). 

*6,6′-bis-(debromo)-gelliusine F* (**1**): brown oil, ^1^H and ^13^C NMR, see Table 2; HRESIMS *m*/*z* 319.2015 [M + H]^+^ (calcd for C_20_H_23_N_4_, 319.1923).

*6-bromo-8,1′-dihydro-isoplysin A* (**2**): yellow oil, $\alpha_{D}^{20}$ 0.0 (*c 0.5*, MeOH); ^1^H and ^13^C NMR, see Table 2; HRESIMS *m*/*z* 337.0483 [M + H]^+^ (calcd for C_14_H_16_N_4_O^81^Br, 337.0487).

*5,6-dibromo-8,1′-dihydro-isoplysin A* (**3**): yellow oil, $\alpha_{D}^{20}$ 0.0 (*c 0.5*, MeOH); ^1^H and ^13^C NMR, see Table 3; HRESIMS *m*/*z* 414.9630 [M + H]^+^ (calcd for C_14_H_15_N_4_O^79^Br^81^Br, 414.9592).

### 3.4. In Vitro Antiplasmodial Assays

Activity against *Plasmodium falciparum* chloroquine-sensitive 3D7 strains was assessed following the procedure already described in Frédérich et al. [18]. The parasites were obtained from MR4-BEI Resources (Manassas, VA, US). Each compound, fraction and extract was applied in a series of eight 2-fold dilutions (final concentrations ranging from 0.8 to 100 μg/mL for an extract and from 0.08 to 10 μg/mL for a pure substance) on two rows of a 96-well microplate and were tested in triplicate (n = 3). Parasite growth was estimated by determination of lactate dehydrogenase activity as described previously [19]. Artemisinin (98%, Sigma-Aldrich, Saint-Louis, MO, USA) was used as positive control with IC_50_ of 0.006 ± 0.002 µg/mL.

### 3.5. In Vitro Antimicrobial Assays

All compounds were tested against five marine bacterial strains commonly found on biofilms, *Roseobacter litoralis* (ATCC 495666)*, Shewanella putrefaciens* (ATCC 8071)*, Vibrio carchariae* (ATCC 35084)*, Vibrio natrigens* (ATCC 14048) and *Vibrio proteolyticus* (ATCC 15338). Bacterial adhesion and growth rates were determined according to the methods of Thabard et al. [20], Messina et al. [21] and Trepos et al. [22]. Bacterial suspensions (100 μ aliquots, 2 × 10^8^ colony forming units/mL) were aseptically added to the microplate wells containing compound (0.01–10 μg/mL), and the plates were incubated for 48 h at 26 °C prior to assessment of bioactivity. Media only (Marine Broth 2216, Difco) was used as a blank. Bacterial growth was monitored spectroscopically at 630 nm. The minimal inhibitory concentration (MIC) for bacterial growth was defined as the lowest concentration which results in a decrease in OD, compared to the blank. The microplates were then emptied, and the bacterial adhesion assay was performed using aqueous crystal staining method [22]. The MIC for bacterial adhesion was defined as the lowest concentration of compound that, after 48-h incubation, produced a decrease of the OD at 595 nm compared to the blank.

## 4. Conclusions

In conclusion, three new tryptophan derived alkaloids, 6,6′-bis-(debromo)-gelliusine F (**1**), 6-bromo-8,1′-dihydro-isoplysin A (**2**) and 5,6-dibromo-8,1′-dihydro-isoplysin A (**3**), were isolated from *Fascaplysinopsis reticulata* together with four known alkaloids from the same family, 8-oxo-tryptamine (**4**), tryptamine (**5**), (*E*)-6-bromo-2′-demethyl-3′-*N*-methylaplysinopsin (**6**) and (*Z*)-6-bromo-2′-demethyl-3′-*N*-methylaplysinopsin (**7**). 6,6′-bis-(debromo)-gelliusine F (**1**) was a new alkaloid with a bis-tryptamine structure and 6-bromo-8,1′-dihydro-isoplysin A (**2**) and 5,6-dibromo-8,1′-dihydro-isoplysin A (**3**) were two new alkaloids with 1′,8-dihydroaplysinopsin structure. The 8-oxo-tryptamine (**4**) and the mixture of the known (*E*) and (*Z*)-6-bromo-2′-demethyl-3′-*N*-methylaplysinopsin (**6**, **7**) exhibited antiplasmodial activity against *Plasmodium falciparum* with IC_50_ values of 8.8 and 8.0 µg/mL respectively while 6,6′-bis-(debromo)-gelliusine F (**1**), 6-bromo-8,1′-dihydro-isoplysin A (**2**), 5,6-dibromo-8,1′-dihydro-isoplysin A (**3**) and tryptamine (**5**) did not show significant antimalarial activity. The two new 6-bromo-8,1′-dihydro-isoplysin A (**2**) and 5,6-dibromo-8,1′-dihydro-isoplysin A (**3**) showed promising antifouling activity against *V. natrigens* and trpyptamine (**5**) showed promising antifouling activity against *V. carchariae*. Further isolation, structure elucidation, and structure-activity relationship studies of this type of alkaloids are required for the development of new drugs.

## Figures and Tables

**Figure 1 marinedrugs-17-00167-f001:**
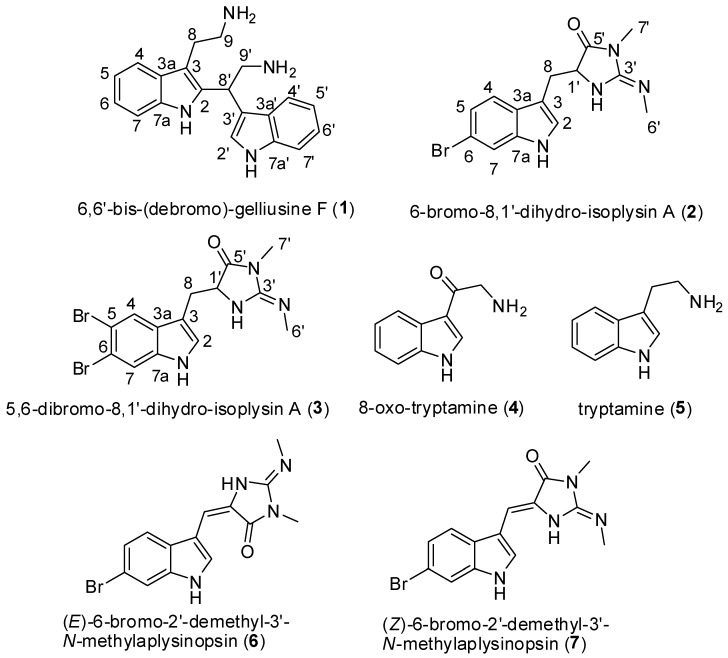
Chemical structures of compounds **1**–**7**.

**Figure 2 marinedrugs-17-00167-f002:**
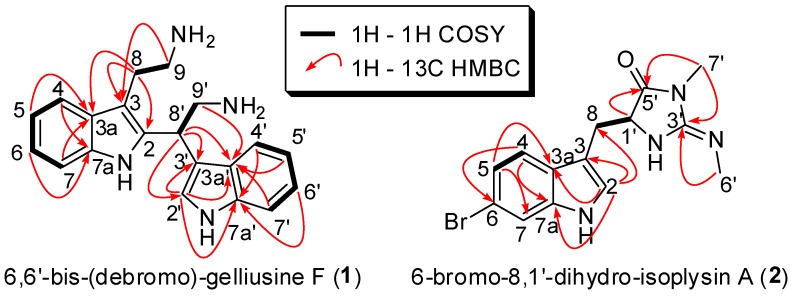
Key COSY and HMBC correlations for compounds **1** and **2.**

**Table 1 marinedrugs-17-00167-t001:** 1D and 2D NMR spectroscopic data (^1^H, ^13^C 300 MHz, CD_3_OD) for 6,6′-bis-(debromo)-gelliusine F (**1**).

Position	δC, Type	δH (*J* in Hz)	COSY (^1^H-^1^H)	HMBC (^1^H-^13^C)
2	124.0, C	-	-	-
3	113.8, C	-	-	-
3a	127.5, C	-	-	-
4	118.9, CH	7.54, d (7.8)	5	6, 7a
5	112.6, CH	7.38, m	4, 6	3a, 7
6	123.1, CH	7.12, m	5, 7	4, 7a
7	120.7, CH	7.06, m	6	3a, 5
7a	135.3, C	-	-	-
8	23.7, CH_2_	3.23, m	9	2, 3, 3a, 9
9	41.4, CH_2_	3.00, m	8	3, 8
2′	124.0, CH	7.27, s	-	3′, 3a’, 7a’
3′	113.7, C	-	-	-
3a’	129.2, C	-	-	-
4′	119.3, CH	7.58, d (7.8)	5′	6′, 7a’
5′	112.9, CH	7.41, m	4′, 6′	3a’, 7′
6′	123.3, CH	7.14, m	5′, 7′	4′, 7a’
7′	120.6, CH	7.06, m	6′	3a’, 5′
7a’	138.0, C	-	-	-
8′	34.3, CH	5.10, t (8.6)	9′	2′, 3′, 3a’, 9′
9′	44.3, CH_2_	3.83–3.69 (m)	8′	3′, 8′

**Table 2 marinedrugs-17-00167-t002:** 1D and 2D NMR spectroscopic data (^1^H, ^13^C 300 MHz, CD_3_OD) for 6-bromo-8,1′-dihydro-isoplysin A (**2**).

Position	δC, Type	δH (*J* in Hz)	COSY (^1^H-^1^H)	HMBC (^1^H-^13^C)
2	126.4, CH	7.11, s	-	3, 3a, 7a, 8
3	109.0, C	-	-	-
3a	127.6, C	-	-	-
4	121.1, CH	7.51, d (8.6)	5	6, 7a
5	123.3, CH	7.14, dd (8.6, 1.8)	4	3a, 7
6	116.6, C	-	-	-
7	115.5, CH	7.50, d (1.8)	-	3a, 5
7a	138.1, C	-	-	-
8	28.1, CH_2_	3.35, m	1′	-
1′	61.8, CH	4.62, t (4.9)	8	5′, 8
3′	159.2, C	-	-	-
5′	174.9, C	-	-	-
6′	25.9, CH_3_	2.90, s	-	3′
7′	29.3, CH_3_	2.86, s	-’	3′, 5′

**Table 3 marinedrugs-17-00167-t003:** Comparison of 1D NMR Spectroscopic Data (^1^H, ^13^C 300 MHz, CD_3_OD for (**2**) and ^1^H 500 MHz, ^13^C 600 MHz, CD_3_OD for (**3**)) between 6-bromo-8,1′-dihydro-isoplysin A (**2**) and 5,6-dibromo-8,1′-dihydro-isoplysin A (**3**).

Position	δH (*J* in Hz)		δC, Type	
	6-Bromo-8,1′-dihydro-isoplysin A (2)	5,6-Dibromo-8,1′-dihydro-isoplysin A (3)	6-Bromo-8,1′-dihydro-isoplysin A (2)	5,6-Dibromo-8,1′-dihydro-isoplysin A (3)
2	7.11, s	7.16, s	126.4, CH	126.6, CH
3	-	-	109.0, C	109.0, C
3a	-	-	127.6, C	129.8, C
4	7.51, d (8.6)	7.96, s	121.1, CH	123.6, CH
5	7.14, dd (8.6, 1.8)	-	123.3, CH	116.9, C
6	-	-	116.6, C	115.9, C
7	7.50, d (1.8)	7.69, s	115.5, CH	117.4, CH
7a	-	-	138.1, C	137.5, C
8	3.35, m	3.73, m	28.1, CH_2_	28.2, CH_2_
1′	4.62, t (4.9)	4.60, t (5.3)	61.8, CH	61.4, CH
3′	-	-	159.2, C	157.9, C
5′	-	-	174.9, C	175.8, C
6′	2.90, s	2.86, s	25.9, CH_3_	25.4, CH_3_
7′	2.86, s	2.94, s	29.3, CH_3_	28.9, CH_3_

**Table 4 marinedrugs-17-00167-t004:** Antimicrobial activities in vitro for pure isolated natural products.

Compounds	*Shewanellia putrefaciens* MIC, µg/mL		*Roseobacter littoralis* MIC, µg/mL		*Vibrio carchariae* MIC, µg/mL		*Vibrio natrigens* MIC, µg/mL		*Vibrio proteolyticus* MIC, µg/mL	
	A	G	A	G	A	G	A	G	A	G
6,6′-bis-(debromo)-gelliusine F (**1**)	-	-	-	-	-	-	-	-	-	-
6-bromo-8,1′-dihydro-isoplysin A (**2**)	-	100	-	-	100	-	100	0.01	-	-
5,6-dibromo-8,1′-dihydro-isoplysin A (**3**)	-	-	-	-	-	-	-	1	-	-
8-oxo-tryptamine (**4**)	-	-	-	-	-	-	-	-	-	-
tryptamine (**5**)	-	-	-	-	-	1	-	-	-	-
(*E*) and (*Z*)-6-bromo-2′-demethyl-3′-*N*-methylaplysinopsine (**6** + **7**)	-	-	-	-	-	-	-	-	-	-

A: Adhesion inhibition; G: Growth inhibition.

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
