# Peer review of "New Antimalarial and Antimicrobial Tryptamine Derivatives from the Marine Sponge Fascaplysinopsis reticulata"

_marinedrugs, 2019, doi:10.3390/md17030167_

Round 1
Reviewer 1 Report
Review concerns Manuscript ID: marinedrugs-460343. Title: New Antimalarial and Antimicrobial Tryptamine Derivatives from the Marine Sponge Fascaplysinopsis reticulate.
In my opinion, a major revision is required before the manuscript can be accepted for publication. The review is focused on structure determination. The remarks are given below.
Structure 1.
Correct data for molecular ion peak according to Figure S1 (The HRESIMS spectrum). It should be 337.0483. Figure S2, 1H NMR spectrum. Integration of the signals is required. Please, correct the data in Table 1 according to the Figure S2 (5.09 not 5.11 (H-8’), 7.14 not 7.15 (H-6’), 7.13 not 7.14 (H-6), 7.57 not 7.55 (H-4). Figure S3, 13C NMR. There is not signal at 113.7 (see Table 1). Please, correct the data in Table 1 according to the Figure S3 (120.7 not 120.9 (C-7), 23.7 not 23.9 (C-8), 41.4 not 41.5 (C-9), 129.2 not 129.1 (C-3a’), 120.6 not 120.7 (C-7’), 138.0 not 137.9 (C-7a’), 34.3 not 34.6 (C-8’), 44.3 not 44.4 (C-9’). Figure S4, COSY – please, make a zoom of the regions at 6.5-8 ppm and 2.5-3.5 ppm. The zooms can be placed in the empty regions of the spectrum, for example, top-left and bottom-right corners. Why 8-9 correlations is not considered in the text? Figure S5, HSQC, should be pointed in the text. Figure S6, HMBC, is not described in the text.
Structure 2.
Figure S7, HRESIMS. The spectrum shows peak at 337.0483 whereas in the main text is 336.3453. Please correct or explain. Figure S8, 1H NMR. Integration of the signals is required. Please, make a zoom at the region 7-8 ppm. Please, mark the exact position of the signal at 3.35 ppm. Please, correct data in Table 2 according to the Figure S8 (7.15 not 7.14 (H-5), 7.50 not 7.49 (H-7), 4.62 not 4.63 (H-1’), 2.90 not 2.91 (H-6’), 2.86 not 2.87 (H-7’). Figure S9, 13C NMR, the NS is too low, so that position of some signals are unclear. Please mark the position of signals at 173.6 (C=O), 157.2 (C=N), 127.1 (C-3a), 107.2 (C-3). There is signal at 115.5 ppm, it is ascribed to C6, C7, or both? In fact the measurement should be repeated, due to the low resolution of 13C NMR, the HMBC spectrum (S12) is also poorly readable. The Figure S13, HSQC, should be marked in the text. Correct the data in Table 2 (28.0 not 26.0 (C-8), correct C-4, C-5, C-1’, C-6’, C-7’).
Structure 3.
Figure S13, HRESIMS is not described in the text. Figure S14, 1H NMR, integration is required. Please, mark position for H-8 (3.75 ppm). Correct data for H-4, H-1’ in Table 3 according to S14.
Author Response
Dear reviewer,
Thank you very much for all your suggestions. Please find in the attached file the answers of your recommandations.
Best regards,

Reviewer 2 Report
The manuscript entitled “New Antimalarial and Antimicrobial Tryptamine Derivatives from the Marine Sponge Fascaplysinopsis reticulate” details the isolation and full characterization of some new natural compounds. Furthermore some biological evaluations were also performed. In my opinion the discussion of the biological results should be improved. The authors should include a section for conclusions.
Author Response
Dear reviewer,
Thank you very much for your recommendation. As suggested, our discussion of the biological results has been improved and a conclusion has been also included.
Best regards,
Round 2
Reviewer 1 Report
Authors have made the corrections, as it was indicated in the previous review. Therefore, in my opinion, the manuscript can be accepted.